# Intramolecular tautomerization of the quercetin molecule due to the proton transfer: QM computational study

**Ol'ha O. Brovarets'[1], Dmytro M. Hovorun[1,2]***

**1** Department of Molecular and Quantum Biophysics, Institute of Molecular Biology and Genetics, National Academy of Sciences of Ukraine, Kyiv, Ukraine, **2** Department of Molecular Biotechnology and Bioinformatics, Institute of High Technologies, Taras Shevchenko National University of Kyiv, Kyiv, Ukraine

* dhovorun@imbg.org.ua

**Data Availability Statement:** All relevant data are within the paper.

**Funding:** The authors received no specific funding for this work.

## Abstract

Quercetin molecule (3, 3′, 4′, 5, 7-pentahydroxyflavone, $C_{15}H_{10}O_7$) is an important flavonoid compound of natural origin, consisting of two aromatic A and B rings linked through the C ring with endocyclic oxygen atom and five hydroxyl groups attached to the 3, 3′, 4′, 5 and 7 positions. This molecule is found in many foods and plants, and is known to have a wide range of therapeutic properties, like an anti-oxidant, anti-toxic, anti-inflammatory etc. In this study for the first time we have revealed and investigated the pathways of the tautomeric transformations for the most stable conformers of the isolated quercetin molecule (Brovarets' & Hovorun, 2019) *via* the intramolecular proton transfer. Energetic, structural, dynamical and polar characteristics of these transitions, in particular relative Gibbs free and electronic energies, characteristics of the intramolecular specific interactions–H-bonds and attractive van der Waals contacts, have been analysed in details. It was demonstrated that the most probable process among all investigated is the proton transfer from the O3H hydroxyl group of the C ring to the C2′ carbon atom of the C2′H group of the B ring along the intramolecular O3H...C2′ H-bond with the further formation of the C2′H$_2$ group. It was established that the proton transfer from the hydroxyl groups to the carbon atoms of the neighboring CH groups is assisted at the transition states by the strong intramolecular HCH...O H-bond (~28.5 kcal·mol$^{-1}$). The least probable path of the proton transfer–from the C8H group to the endocyclic O1 oxygen atom–causes the decyclization of the C ring in some cases. It is shortly discussed the biological importance of the obtained results.

## Introduction

Quercetin molecule is a compound of natural origin, which is found in different foods and plants [1, 2]. This compound has attracted a lot of attention, since it is suggested to have a wide range of properties, such as an anti-oxidant, anti-toxic, anti-inflammatory etc. [3–20]. The structure of the quercetin contains two (A+C) and B rings and also has five hydroxyl groups at the 3, 3′, 4′, 5, 7 positions. So, due to these structural features it can acquire different

**Competing interests:** The authors have declared that no competing interests exist.

conformations [21–26] and perform structural transitions between them due to the mutual rotations of the (A+C) and B rings around the C2-C1′ bond and also of the hydroxyl groups OH around the exocyclic C-O bonds [27]. Thus, investigations of the conformational transformations through the rotations around the C2-C1′ bond have been presented in literature [28–30]. In particular, in our recent works we have investigated in details conformational variety [26] and also conformational transitions of the quercetin molecule *via* the rotations of its rings [31] by using the quantum-mechanical (QM) calculations at the MP2/6-311++G(d,p)// B3LYP/6-311++G(d,p) level of theory and Bader's quantum theory of "Atoms in Molecules" (QTAIM). Altogether, as a result of the study it was revealed 48 stable conformers (24 planar and 24 non-planar) with relative Gibbs free energies within the range of 0.0–25.3 kcal·mol$^{-1}$ under normal conditions, stabilized by the H-bonds (both classical OH. . .O and so-called unusual CH. . .O and OH. . .C) and attractive van der Waals contacts O. . .O, which have been divided into four different subfamilies by their structural properties [26]. Conformers of the quercetin molecule have been established to be polar structures with a dipole moment, which varies within the range from 0.35 to 9.87 Debay. We have also found out the interconversions of the 24 pairs of the conformers of the quercetin molecule *via* the rotation of its practicallay non-deformable (A+C) and B rings around the C2-C1' bond through the quasi-orthogonal transition states with Gibbs free energies of activation in the range of 2.17–5.68 kcal·mol$^{-1}$ at normal conditions [31]. It was also provided comprehensive analysis of the 123 prototropic tautomers of the quercetin molecule [32, 33].

Also, quercetin can potentially acquire different prototropic tautomeric forms, but these data are weakly presented in the literature yet [34–38], despite the continuous comprehensive research of the quercetin molecule during the last decades [21–38].

It is widely known from the literature data that *proton transfer* is important biochemical phenomenon and plays important role in the biochemical reactions [39–45]. Thus, it was established that even movement of the single proton (SP) from the one site to another can cause significant changes of the energetic, structural and dynamical properties of the molecule, thus changing its functionality. In particular, tautomerization *via* the single (SPT) or double (DPT) proton transfer has been established for the canonical or non-canonical DNA base pairs [46–55], by the participation as of classical DNA bases [56], so by the participation of modified bases such as hypoxanthine [57–59], 5-bromouracil [60] and 2-aminopurine [61–64] molecules.

Currently, there are only some studies in the literature, devoted to the prototropic tautomerism of the quercetin molecule, in particular keto-enol tautomerism [34–38]. Their importance is caused by the relevance of the quercetin tautomers to the hydrogen↔deuterium (H↔D) exchange processes of its CH-groups [36], irreversible structural changes of the quercetin molecule at the increasing of the temperature [34–35] and tautomerization of the quercetin molecule at the transition to an excited electronic state [38]. However, possible ways of the formation of the rare prototropic tautomers of the quercetin molecule have not been carefully considered.

So, the aim of this study is to reveal and investigate the possible pathways of the prototropic transformations of the isolated quercetin molecule [65].

As a result of this scrupulous investigation we have revealed possible ways of tautomerization of the quercetin molecule *via* the single proton transfer, which are entangled with the following phenomena (Fig 1):

a. Proton transfer from the C8H group to the neighboring O1 oxygen atom.

b. Transition of the proton from the O7H/O3′H hydroxyl groups to the carbon atoms of the neighboring C6H/C2′H groups.

## (a) Transition of the proton from the C8H group to the neighboring O1 oxygen atom

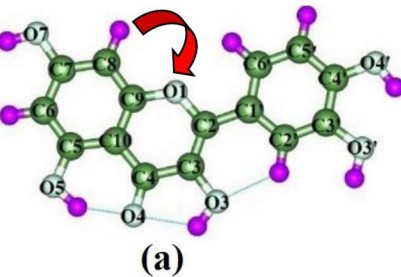

(a)

## (b), (c) Transition of the proton from the O7H/O3′H hydoxyl groups to the neighboring C6/C2′ carbon atoms of the C6H/C2′H groups and from the O7H/O5H/O3H/O4′H hydoxyl groups to the C8/C6/C2′/C5′ carbon atoms of the C8H/C6H/C2′H/C5′H groups, preceded by the rotations of the hydroxyl groups around the C7O7/C5O5/C3O3/C4′O4′ bonds by 180 degree

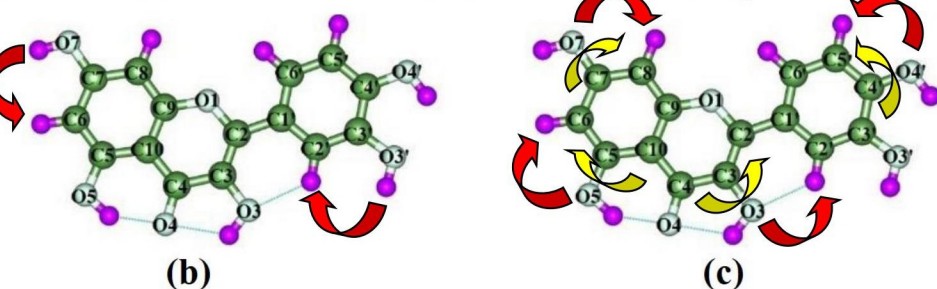

(b)                                          (c)

## (d), (e) Transition of the proton from the O3H hydroxyl group to the neighboring O4 oxygen atom and from the O7H/O5H hydroxyl group to the C6 carbon atom of the C6H group

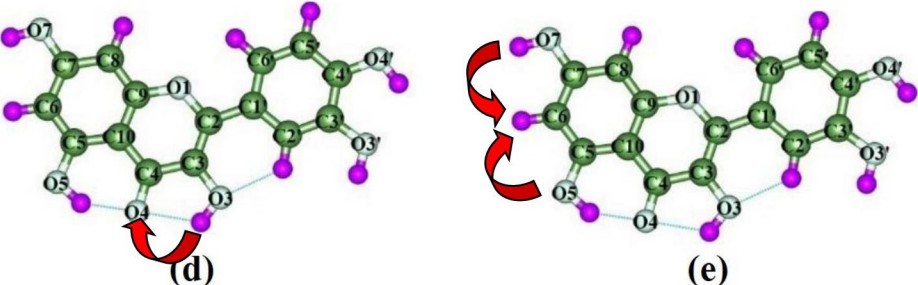

(d)                                          (e)

**Fig 1. Schematic representations of the possible mechanisms of the intramolecular proton mobility in the quercetin molecule.** Red arrows denote the directions of the proton transfer, yellow–rotations of the hydroxyl groups around the C-O bond by 180 degree.

  c. Migration of the proton from the O7H/O5H/O3H/O4′H hydroxyl groups to the carbon atoms of the neighboring C8H/C6H/C2′H/C5′H groups, preceded by the rotations of the hydroxyl groups around the C7O7/C5O5/C3O3/C4′O4′ bonds by 180 degree.

  d. Proton transfer from the O3H hydroxyl group to the neighboring O4 oxygen atom.

  e. Transition of the proton from the O7H/O5H hydroxyl groups to the C6 carbon atom of the neighboring C6H group.

Number of the important physico-chemical parameters of these transformations, in particular relative Gibbs free and electronic energies, characteristics of the intramolecular H-bonds and attractive van der Waals interactions have been analysed in details. Especial attention has been focused on the processes of the intramolecular tautomerization by proton transfer, which are more or less likely to occur. Possible chemical and biological roles of the obtained results have been shortly outlined.

## Computational methods

We have used the DFT B3LYP/6-311++G(d,p) level of theory [66–69], incorporated into Gaussian'09 program package [70], to provide the calculations of the geometrical structures and vibrational spectra of the prototropic tautomers of the quercetin molecule and transitions states (TSs) between them, which have been localized by Synchronous Transit-guided Quasi-Newton method [66]. This level of theory has been successfully approved for the calculations of the heterocyclic compounds [71–78]. Scaling factor of 0.9668 has been applied for the correction of the harmonic frequencies for the investigated structures [79, 80]. Electronic and Gibbs free energies under normal conditions have been calculated by single point calculations at the MP2/6-311++G(2df,pd) level of theory [81–83].

The Hessian-based predictor-corrector integration algorithm [84] has been applied for obtaining the IRC pathways in the forward and reverse directions from each TS.

The time $\tau_{99.9\%}$, which is necessary to reach 99.9% of the equilibrium concentration of the reactant and product, the lifetime $\tau$ $(1/k_r)$ of the prototropic tautomers, the forward $k_f$ and reverse $k_r$ rate constants have been obtained by the well-known formulas of physico-chemical kinetics [85], respectively:

$$\tau_{99.9\%} = \frac{ln10^3}{k_f + k_r} \tag{1}$$

$$k_{f,r} = \Gamma \cdot \frac{k_B T}{h} e^{-\frac{\Delta\Delta G_{f,r}}{RT}} \tag{2}$$

where quantum tunneling effect has been accounted by Wigner's tunneling correction $\Gamma$ [86–88]:

$$\Gamma = 1 + \frac{1}{24} \left( \frac{h\nu_i}{k_B T} \right)^2 \tag{3}$$

where $k_B$–Boltzmann's constant, $h$–Planck's constant, $\Delta\Delta G_{f,r}$–Gibbs free energy of activation for the conformational transition in the forward ($f$) and reverse ($r$) directions, $\nu_i$–magnitude of the imaginary frequency associated with the vibrational mode at the TSs.

The distribution of the electron density has been analyzed by application of the program package AIM'2000 [89] with all default options and wave functions obtained at the B3LYP/6-311++G(d,p) level of theory for geometry optimisation. The presence of the (3,-1) bond critical point (BCP), bond path between hydrogen donor and acceptor and positive value of the Laplacian at this BCP ($\Delta\rho>0$) have been considered as criteria for the formation of the H-bond and attractive van der Waals contact [62–63, 90–92].

Energies of the unusual intramolecular CH···O and OH···C H-bonds and attractive O···O and C···O van der Waals contacts have been obtained using Bader's quantum theory of Atoms in Molecules [93] by the empirical Espinosa-Molins-Lecomte (EML) formula [94, 95], based on the electron density distribution at the (3,-1) BCPs of the H-bonds:

$$E_{CH\cdots O/OH\cdots C/O\cdots O/C\cdots O} = 0.5 \cdot V(r) \tag{4}$$

where V(r)–value of a local potential energy at the (3,-1) BCP.

It should be noted, that for the CH...O H-bonds, which are strong with energy that exceeds 10 kcal·mol⁻¹, their energy have been estimated by the Brovarets'-Yurenko-Hovorun formula [96, 97], considered in the literature [98, 99]:

$$E_{CH\cdots O} = 248.501 \cdot \rho - 0.367 \tag{5}$$

The energies of the classical intramolecular OH···O H-bonds have been calculated by the Nikolaienko-Bulavin-Hovorun formula [100]:

$$E_{OH\cdots O} = -3.09 + 239 \cdot \rho \tag{6}$$

where ρ–the electron density at the (3,-1) BCP of the H-bond.

All calculations have been performed for the tautomeric transitions of the quercetin molecule as their intrinsic property, that is adequate for modeling of the processes occurring in real systems [101–106].

In this work standard numeration of atoms has been used [2]. At this, prototropic tautomers of the quercetin molecule have been designated by the asterisk; subscript corresponds to the localization of the mobile protons. Numeration of the conformers (highlighted in bold) is the same, as in the previous work [26].

## Results and discussion

In the process of this study we have suggested different ways of the formation of the prototropic tautomers of the most stable conformer **1** [26] of the quercetin molecule. Then, by using the method of "trials and errors" we have localized TSs for these tautomeric transformations, occurring *via* the intramolecular proton transfer. However, only some of the suggested tautomeric transformations have been confirmed, while others of them have been modified in the course of the investigation.

So, in this study we have considered the following mechanisms of the tautomerization of the quercetin molecule, in particular of the most stable conformer **1**, that can proceed in the different ways through the intramolecular proton transfer (see Figs 1 and 2, Tables 1 and 2).

It was established that these transformations of the quercetin molecule are accompanied by the changes of their geometry, dipole moment rearrangement and breakage and formation of the intramolecular specific contacts (H-bonds and attractive van der Waals contacts).

Analysis of the investigated mechanisms and their discussion are provided further one-by-one.

**a) Proton transfer from the C8H group to the O1 atom.** First of the considered mechanisms consists in the intramolecular transition of the proton, localized at the C8 carbon atom, to the neighboring endocyclic oxygen atom O1, leading to the formation of the new tautomer with formed O1H hydroxyl group (Figs 1 and 2). We have analysed this tranformation for the case of the main stable conformer **1** of the quercetin molecule and also checked it for the others–conformers **4**, **7** and **10**: $\mathbf{1}\leftrightarrow\mathbf{1^*_{O1H}}$ ($\Delta\Delta G_{TS} = 96.31$); $\mathbf{4}\leftrightarrow\mathbf{4^*_{O1H}}$ ($\Delta\Delta G_{TS} = 92.66$); $\mathbf{7}\leftrightarrow\mathbf{7^*_{O1H}}$ ($\Delta\Delta G_{TS} = 96.26$) and $\mathbf{10}\leftrightarrow\mathbf{10^*_{O1H}}$ ($\Delta\Delta G_{TS} = 96.27$ kcal·mol⁻¹) (Table 1).

Finally, four new prototropic tautomers have been formed– $\mathbf{1^*_{O1H}}$ ($\Delta G = 9.20$), $\mathbf{4^*_{O1H}}$ ($\Delta G = 0.90$), $\mathbf{7^*_{O1H}}$ ($\Delta G = 9.03$) and $\mathbf{10^*_{O1H}}$ ($\Delta G = 9.14$ kcal·mol⁻¹) (Table 1). Notably, all of them, except the case of the conformer **4**, which contains opened C-ring and new exotic strong attractive van der Waals contact C9 o...O1 (~6.5 kcal·mol⁻¹ (Table 2)) instead of the C9-O1 covalent bond in the C ring. In the case of the $\mathbf{4}\leftrightarrow\mathbf{4^*_{O1H}}$ tautomeric transition, the covalent bond C9-O1 survives during this transformation. Notably, three lower H-bonds, stabilizing conformers–O5H...O4, O3H...O4 and C2'H...O3,–remain the same, changing only their energies during tautomerisation (Table 2).

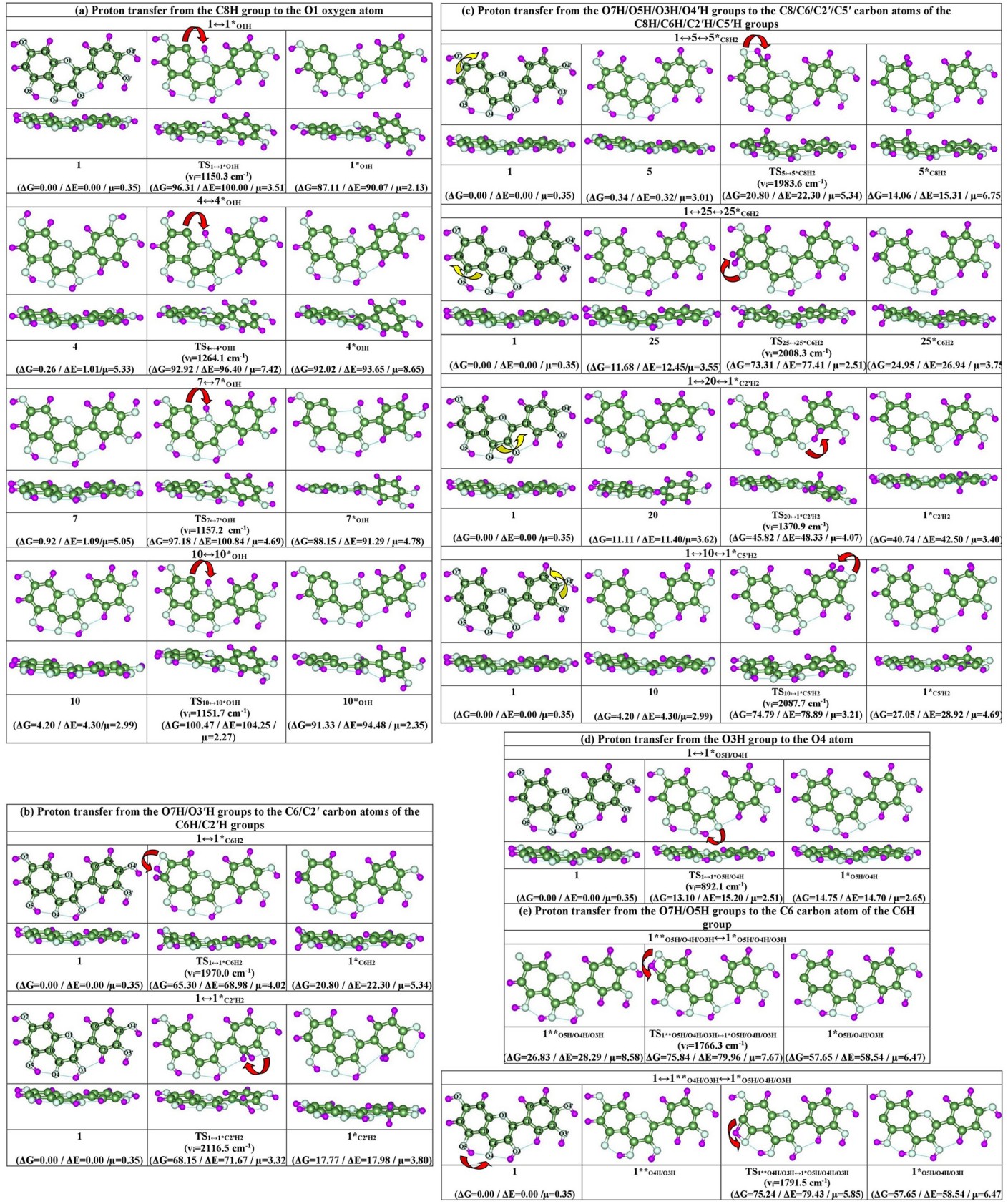

**Fig 2. Reaction pathways for the intramolecular proton transfer in the isolated quercetin molecule; initial and terminal states with TSs between them have been obtained at the MP2/6-311++G(2df,pd) // B3LYP/6-311++G(d,p) level of QM theory (low index near formed tautomers denotes the site of the localization of the transferred proton).** Gibbs free ΔG and electronic ΔE energies (kcal·mol⁻¹), imaginary frequencies $v_i$ at the TS and dipole moments μ (Debay) are provided below reaction paths. Dotted lines indicate intramolecular specific interactions. Red arrows denote the intramolecular transition of the proton, while yellow arrows– rotations of the hydroxyl groups. See also Tables 1 and 2.

The $1 \leftrightarrow 1^*_{O1H}$ tautomerisation reaction occurs *via* quite high activation barrier and $TS_{1 \leftrightarrow 1^* O1H}$ with high imaginary frequency ($v_i$ = 1150.3 cm⁻¹). Notably, that we have checked and revealed that this transition is typical for all investigated conformers. Thus, the Gibbs free

**Table 1. Energetic and kinetic characteristics of the tautomeric transformations by the intramolecular proton transfer in the isolated quercetin molecule obtained at the MP2/6-311++G(2df,pd)//B3LYP/6-311++G(d,p) level of QM theory under normal conditions (see also Fig 1).**

| Tautomeric transition | $v_i$[a] | $\Delta G$[b] | $\Delta E$[c] | $\Delta\Delta G_{TS}$[d] | $\Delta\Delta E_{TS}$[e] | $\Delta\Delta G$[f] | $\Delta\Delta E$[g] | $k_f$[h] | $k_r$[i] | $\tau_{99.9\%}$[j] | $\tau$[k] |
|---|---|---|---|---|---|---|---|---|---|---|---|
| (a) Proton transfer from the C8H group to the O1 oxygen atom | | | | | | | | | | | |
| $1 \leftrightarrow 1^*_{O1H}$ | 1150.3 | 87.11 | 90.07 | 96.31 | 100.00 | 9.20 | 9.93 | $3.09 \cdot 10^{-58}$ | $2.43 \cdot 10^{6}$ | $2.84 \cdot 10^{-6}$ | $4.12 \cdot 10^{-7}$ |
| $4 \leftrightarrow 4^*_{O1H}$ | 1264.1 | 91.76 | 92.64 | 92.66 | 95.39 | 0.90 | 2.75 | $1.64 \cdot 10^{-55}$ | $3.31 \cdot 10^{12}$ | $2.09 \cdot 10^{-12}$ | $3.02 \cdot 10^{-13}$ |
| $7 \leftrightarrow 7^*_{O1H}$ | 1157.2 | 87.23 | 90.20 | 96.26 | 99.75 | 9.03 | 9.55 | $3.58 \cdot 10^{-58}$ | $3.26 \cdot 10^{6}$ | $2.12 \cdot 10^{-6}$ | $3.07 \cdot 10^{-7}$ |
| $10 \leftrightarrow 10^*_{O1H}$ | 1151.7 | 87.13 | 90.18 | 96.27 | 99.95 | 9.14 | 9.77 | $3.31 \cdot 10^{-58}$ | $2.69 \cdot 10^{6}$ | $2.57 \cdot 10^{-6}$ | $3.72 \cdot 10^{-7}$ |
| (b) Proton transfer from the O7H/O3′H groups to the C6/C2′ carbon atoms of the C6H/C2′H groups | | | | | | | | | | | |
| $1 \leftrightarrow 1^*_{C6H2}$ | 1970.0 | 20.80 | 22.30 | 65.30 | 68.98 | 44.50 | 46.68 | $3.53 \cdot 10^{-35}$ | $6.37 \cdot 10^{-20}$ | $1.08 \cdot 10^{20}$ | $1.57 \cdot 10^{19}$ |
| $1 \leftrightarrow 1^*_{C2'H2}$ | 2116.5 | 17.77 | 17.98 | 68.15 | 71.67 | 50.38 | 53.69 | $3.21 \cdot 10^{-37}$ | $3.47 \cdot 10^{-24}$ | $1.99 \cdot 10^{24}$ | $2.88 \cdot 10^{23}$ |
| (c) Proton transfer from the O7H/O5H/O3H/O4′H groups to the C8/C6/C2′/C5′ carbon atoms of the C8H/C6H/C2′H/C5′H groups | | | | | | | | | | | |
| $5 \leftrightarrow 5^*_{C8H2}$ | 1983.6 | 13.72 | 14.99 | 20.46 | 21.98 | 6.74 | 6.99 | $2.77 \cdot 10^{-2}$ | $3.21 \cdot 10^{8}$ | $2.15 \cdot 10^{-8}$ | $3.12 \cdot 10^{-9}$ |
| $25 \leftrightarrow 25^*_{C6H2}$ | 2008.3 | 13.27 | 14.49 | 61.63 | 64.96 | 48.36 | 50.47 | $1.79 \cdot 10^{-32}$ | $9.69 \cdot 10^{-23}$ | $7.13 \cdot 10^{22}$ | $1.03 \cdot 10^{22}$ |
| $20 \leftrightarrow 1^{**}_{C2'H2}$ | 1370.9 | 29.63 | 31.10 | 34.71 | 36.93 | 5.08 | 5.83 | $5.79 \cdot 10^{-13}$ | $3.14 \cdot 10^{9}$ | $2.20 \cdot 10^{-9}$ | $3.19 \cdot 10^{-10}$ |
| $10 \leftrightarrow 1^*_{C5'H2}$ | 2087.7 | 22.85 | 24.62 | 70.59 | 74.59 | 47.74 | 49.97 | $5.09 \cdot 10^{-39}$ | $2.93 \cdot 10^{-22}$ | $2.35 \cdot 10^{22}$ | $3.41 \cdot 10^{21}$ |
| (d) Proton transfer from the O3H group to the O4 atom | | | | | | | | | | | |
| $1 \leftrightarrow 1^*_{O5H/O4H}$ | 892.1 | 14.30 | 14.70 | 13.10 | 15.20 | -1.20 | 0.50 | $2.62 \cdot 10^{3}$ | $8.09 \cdot 10^{13}$ | $8.54 \cdot 10^{-14}$ | $1.24 \cdot 10^{-14}$ |
| (e) Proton transfer from the O7H/O5H groups to the C6 carbon atom of the C6H group | | | | | | | | | | | |
| $1^{**}_{O5H/O4H/O3H} \leftrightarrow 1^*_{O5H/O4H/O3H}$ | 1766.3 | 30.82 | 30.25 | 49.01 | 51.67 | 18.19 | 21.42 | $2.66 \cdot 10^{-23}$ | 1.07 | 6.43 | 0.93 |
| $1^{**}_{O4H/O3H} \leftrightarrow 1^*_{O5H/O4H/O3H}$ | 1791.5 | 57.65 | 58.54 | 75.24 | 79.43 | 17.59 | 20.89 | $1.56 \cdot 10^{-42}$ | 3.02 | 2.29 | 0.33 |

[a]The imaginary frequency at the TS of the tautomeric transition, cm⁻¹.

[b]The Gibbs free energy of the initial relatively the terminal structure of the tautomerisation reaction (T = 298.15 K), kcal·mol⁻¹.

[c]The electronic energy of the initial relatively the terminal structure of the tautomerisation reaction, kcal·mol⁻¹.

[d]The Gibbs free energy barrier for the forward tautomerisation reaction, kcal·mol⁻¹.

[e]The electronic energy barrier for the forward tautomerisation reaction, kcal·mol⁻¹.

[f]The Gibbs free energy barrier for the reverse tautomerisation reaction, kcal·mol⁻¹.

[g]The electronic energy barrier for the reverse tautomerisation reaction, kcal·mol⁻¹.

[h]The rate constant for the forward tautomerisation reaction, s⁻¹.

[i]The rate constant for the reverse tautomerisation reaction, s⁻¹.

[j]The time necessary to reach 99.9% of the equilibrium concentration between the reactant and the product of the tautomerisation reaction, s.

[k]The lifetime of the product of the tautomerisation reaction, s.

**Table 2. Energetical, electron-topological and geometrical characteristics of the intramolecular specific contacts–H-bonds and attractive van der Waals (vdW) contacts, polar and geometrical parameters of the investigated structures of the isolated quercetin molecule obtained at the B3LYP/6-311++G(d,p) level of QM theory (see also Fig 1).**

| Conformer, TS and tautomer | AH···B H-bond / A···B vdW contact | $E_{AH\cdots B} / E_{A\cdots B}$ [a] | $\rho$ [b] | $\Delta\rho$ [c] | $100\cdot\varepsilon$ [d] | $d_{A\cdots B}$ [e] | $d_{H\cdots B}$ [f] | $\angle AH\cdots B$ [g] | $\Delta C3\text{-}C2\text{-}C1'\text{-}C6'$ [h] | $\mu$ [i] |
|---|---|---|---|---|---|---|---|---|---|---|
| **(a) Transition of the proton from the C8H group to the O1 oxygen atom** | | | | | | | | | | |
| $1\leftrightarrow1^*_{O1H}$ | | | | | | | | | | |
| **1** | O5H...O4 | 6.71 | 0.041 | 0.124 | 1.52 | 2.655 | 1.770 | 147.3 | 180.0 | 0.35 |
| | O3H...O4 | 3.36 | 0.027 | 0.103 | 60.55 | 2.625 | 2.009 | 119.0 | | |
| | C2'H...O3 | 4.01* | 0.018 | 0.076 | 0.92 | 2.883 | 2.137 | 123.8 | | |
| **TS$_{1\leftrightarrow1^*O1H}$** | O5H...O4 | 2.89 | 0.025 | 0.082 | 1.10 | 2.830 | 1.990 | 142.9 | -164.8 | 3.38 |
| | O3H...O4 | 3.84 | 0.029 | 0.109 | 38.00 | 2.600 | 1.963 | 120.4 | | |
| | C2'H...O3 | 3.32* | 0.016 | 0.063 | 10.19 | 2.924 | 2.221 | 120.6 | | |
| **1$^*_{O1H}$ [32]** | O5H...O4 | 9.10 | 0.051 | 0.148 | 2.33 | 2.565 | 1.672 | 147.8 | -155.3 | 1.83 |
| | O3H...O4 | 5.75 | 0.037 | 0.134 | 17.66 | 2.509 | 1.853 | 121.6 | | |
| | C9...O1 | 6.47* | 0.029 | 0.090 | 2.46 | 2.895 | - | - | | |
| | C2'H...O3 | 3.38* | 0.016 | 0.064 | 26.96 | 2.856 | 2.253 | 113.1 | | |
| $4\leftrightarrow4^*_{O1H}$ | | | | | | | | | | |
| **4** | O5H...O4 | 6.47 | 0.040 | 0.124 | 1.46 | 2.659 | 1.776 | 147.3 | 180.0 | 5.33 |
| | O3H...O4 | 3.36 | 0.027 | 0.103 | 60.63 | 2.624 | 2.009 | 118.9 | | |
| | C6'H...O3 | 3.83* | 0.018 | 0.073 | 0.80 | 2.895 | 2.159 | 123.3 | | |
| **TS$_{4\leftrightarrow4^*O1H}$** | O5H...O4 | 2.65 | 0.024 | 0.081 | 1.20 | 2.835 | 1.995 | 143.0 | 162.2 | 7.77 |
| | O3H...O4 | 3.84 | 0.029 | 0.109 | 38.52 | 2.601 | 1.965 | 120.2 | | |
| | C6'H...O3 | 3.07* | 0.015 | 0.058 | 12.45 | 2.942 | 2.261 | 119.2 | | |
| **4$^*_{O1H}$ [32]** | O5H...O4 | 5.04 | 0.034 | 0.112 | 1.32 | 2.705 | 1.842 | 145.2 | 154.4 | 8.88 |
| | O3H...O4 | 4.32 | 0.031 | 0.113 | 30.34 | 2.579 | 1.939 | 120.5 | | |
| | C6'H...O3 | 2.57* | 0.012 | 0.048 | 22.18 | 2.975 | 2.358 | 114.7 | | |
| $7\leftrightarrow7^*_{O1H}$ | | | | | | | | | | |
| **7** | O5H...O4 | 6.71 | 0.041 | 0.125 | 1.50 | 2.652 | 1.767 | 147.3 | 180.0 | 5.05 |
| | O3H...O4 | 3.12 | 0.026 | 0.102 | 68.01 | 2.630 | 2.020 | 118.5 | | |
| | C2'H...O3 | 3.83* | 0.018 | 0.073 | 0.02 | 2.886 | 2.160 | 122.4 | | |
| **TS$_{7\leftrightarrow7^*O1H}$** | O5H...O4 | 2.89 | 0.025 | 0.082 | 1.09 | 2.828 | 1.988 | 142.9 | -162.0 | 4.84 |
| | O3H...O4 | 3.60 | 0.028 | 0.107 | 42.14 | 2.606 | 1.977 | 119.8 | | |
| | C2'H...O3 | 3.03* | 0.014 | 0.058 | 13.43 | 2.936 | 2.267 | 118.2 | | |
| **7$^*_{O1H}$ [32]** | O5H...O4 | 9.10 | 0.051 | 0.147 | 2.27 | 2.565 | 1.673 | 147.8 | -151.5 | 4.65 |
| | O3H...O4 | 5.28 | 0.035 | 0.131 | 19.50 | 2.516 | 1.869 | 121.0 | | |
| | C9...O1 | 6.54* | 0.029 | 0.091 | 2.45 | 2.374 | - | - | | |
| | C2'H...O3 | 2.98* | 0.014 | 0.056 | 36.27 | 2.876 | 2.327 | 109.7 | | |
| $10\leftrightarrow10^*_{O1H}$ | | | | | | | | | | |
| **10** | O5H...O4 | 6.71 | 0.041 | 0.124 | 1.51 | 2.654 | 1.770 | 147.3 | 180.0 | 2.99 |
| | O3H...O4 | 3.12 | 0.026 | 0.103 | 61.90 | 2.626 | 2.011 | 118.9 | | |
| | C2'H...O3 | 3.98* | 0.018 | 0.075 | 1.02 | 2.889 | 2.141 | 124.1 | | |
| **TS$_{10\leftrightarrow10^*O1H}$** | O5H...O4 | 2.89 | 0.025 | 0.082 | 1.09 | 2.830 | 1.990 | 142.9 | -164.7 | 2.34 |
| | O3H...O4 | 3.84 | 0.029 | 0.109 | 38.76 | 2.601 | 1.966 | 120.3 | | |
| | C2'H...O3 | 3.27* | 0.015 | 0.062 | 10.11 | 2.932 | 2.227 | 120.8 | | |
| **10$^*_{O1H}$ [32]** | O5H...O4 | 9.10 | 0.051 | 0.148 | 2.31 | 2.565 | 1.673 | 147.8 | -154.8 | 2.24 |
| | O3H...O4 | 5.51 | 0.036 | 0.133 | 18.06 | 2.510 | 1.857 | 121.5 | | |
| | C9...O1 | 6.52* | 0.029 | 0.091 | 2.51 | 2.375 | - | - | | |
| | C2'H...O3 | 3.30* | 0.015 | 0.062 | 27.26 | 2.865 | 2.263 | 113.1 | | |

*(Continued)*

**Table 2.** (*Continued*)

| Conformer, TS and tautomer | AH···B H-bond / A···B vdW contact | $E_{AH···B} / E_{A··B}$ [a] | $\rho$ [b] | $\Delta\rho$ [c] | $100·\varepsilon$ [d] | $d_{A···B}$ [e] | $d_{H···B}$ [f] | $\angle AH···B$ [g] | $\Delta C3\text{-}C2\text{-}C1'\text{-}C6'$ [h] | $\mu$ [i] |
|---|---|---|---|---|---|---|---|---|---|---|
| **(b) Transition of the proton from the O7H/O3′H groups to the neighboring C6/C2′ carbon atoms of the C6H/C2′H groups** | | | | | | | | | | |
| **1↔1\*$_{C6H2}$** | | | | | | | | | | |
| **TS$_{1↔1^*C6H2}$** | HC6H . . .O7 | 28.46\*\* | 0.116 | 0.059 | 20.81 | 2.211 | 1.388 | 105.3 | 179.3 | 3.96 |
| | O5H . . .O4 | 8.62 | 0.049 | 0.136 | 1.51 | 2.595 | 1.692 | 148.4 | | |
| | O3H . . .O4 | 3.12 | 0.026 | 0.101 | 69.93 | 2.633 | 2.025 | 118.4 | | |
| | C2'H . . .O3 | 3.97\* | 0.018 | 0.075 | 1.12 | 2.886 | 2.141 | 123.8 | | |
| **1\*$_{C6H2}$** [32] | O5H . . .O4 | 8.86 | 0.050 | 0.137 | 1.47 | 2.587 | 1.687 | 147.5 | 180.0 | 5.34 |
| | O3H . . .O4 | 2.65 | 0.024 | 0.099 | 96.94 | 2.646 | 2.048 | 117.7 | | |
| | C2'H . . .O3 | 4.14\* | 0.019 | 0.078 | 1.09 | 2.872 | 2.126 | 123.9 | | |
| **1↔1\*$_{C2'H2}$** | | | | | | | | | | |
| **TS$_{1↔1^*C2'H2}$** | O5H . . .O4 | 6.47 | 0.040 | 0.124 | 1.37 | 2.657 | 1.775 | 147.0 | -176.8 | 3.94 |
| | O3H . . .O4 | 2.89 | 0.025 | 0.100 | 76.39 | 2.637 | 2.032 | 118.1 | | |
| | C2'H . . .O3 | 3.05\* | 0.014 | 0.058 | 39.78 | 2.833 | 2.322 | 106.5 | | |
| | HC2'H . . .O3' | 26.97\*\* | 0.110 | 0.072 | 28.82 | 2.244 | 1.409 | 105.0 | | |
| **1\*$_{C2'H2}$** [32] | O5H . . .O4 | 6.47 | 0.040 | 0.124 | 1.43 | 2.657 | 1.775 | 147.1 | 180.0 | 3.80 |
| | O3H . . .O4 | 3.12 | 0.026 | 0.102 | 68.85 | 2.631 | 2.022 | 118.4 | | |
| | O3 . . .C2' | 2.90\* | 0.012 | 0.056 | 374.04 | 2.807 | - | - | | |
| | O4'H . . .O3' | 2.65 | 0.024 | 0.098 | 139.26 | 2.650 | 2.071 | 116.2 | | |
| **(c) Transition of the proton from the O7H/O5H/O3H/O4′H groups to the carbon atoms of the C8H/C6H/C2′H/C5′H groups, preceded by the rotations of the hydroxyl groups around the C7O7/C5O5/C3O3/C4′O4′ axes by 180 degree** | | | | | | | | | | |
| **5↔5\*$_{C8H2}$** | | | | | | | | | | |
| **5** | O5H . . .O4 | 6.47 | 0.040 | 0.123 | 1.47 | 2.660 | 1.777 | 147.2 | 180.0 | 3.01 |
| | O3H . . .O4 | 3.36 | 0.027 | 0.104 | 58.22 | 2.623 | 2.004 | 119.1 | | |
| | C2'H . . .O3 | 4.01\* | 0.018 | 0.076 | 0.91 | 2.883 | 2.138 | 123.8 | | |
| **TS$_{5↔5^*C8H2}$** | O5H . . .O4 | 8.35 | 0.048 | 0.136 | 1.28 | 2.607 | 1.700 | 149.6 | 177.3 | 4.60 |
| | O3H . . .O4 | 6.78 | 0.026 | 0.103 | 61.23 | 2.623 | 2.010 | 118.6 | | |
| | C2'H . . .O3 | 3.87\* | 0.018 | 0.073 | 1.71 | 2.894 | 2.151 | 123.6 | | |
| | HC8H . . .O7 | 30.20\*\* | 0.123 | 0.033 | 17.10 | 2.209 | 1.363 | 105.4 | | |
| **5\*$_{C8H2}$** [32] | O5H . . .O4 | 7.19 | 0.043 | 0.130 | 1.07 | 2.642 | 1.746 | 149.1 | 180.0 | 6.69 |
| | O3H . . .O4 | 3.36 | 0.027 | 0.104 | 62.47 | 2.620 | 2.008 | 118.5 | | |
| | C2'H . . .O3 | 3.84\* | 0.018 | 0.073 | 0.71 | 2.897 | 2.154 | 123.7 | | |
| **25↔25\*$_{C6H2}$** | | | | | | | | | | |
| **25** | O3H . . .O4 | 4.56 | 0.032 | 0.117 | 31.44 | 2.571 | 1.921 | 121.2 | 180.0 | 3.55 |
| | O5 . . .O4 | 2.91\* | 0.012 | 0.049 | 13.37 | 2.765 | - | - | | |
| | C2'H . . .O3 | 3.93\* | 0.018 | 0.074 | 0.62 | 2.890 | 2.145 | 123.8 | | |
| **TS$_{25↔25^*C6H2}$** | HC6H . . .O5 | 31.44\*\* | 0.128 | 0.012 | 15.09 | 2.208 | 1.347 | 105.3 | 179.7 | 2.51 |
| | O3H . . .O4 | 4.16 | 0.030 | 0.112 | 2.00 | 2.590 | 1.950 | 120.5 | | |
| | C2'H . . .O3 | 3.87\* | 0.018 | 0.073 | 0.56 | 2.895 | 2.151 | 123.8 | | |
| **25\*$_{C6H2}$** [32] | O5 . . .O4 | 2.55\* | 0.010 | 0.041 | 188.30 | 2.903 | - | - | 179.6 | 4.46 |
| | O3H . . .O4 | 4.80 | 0.033 | 0.118 | 28.83 | 2.564 | 1.909 | 121.5 | | |
| | C2'H . . .O3 | 3.75\* | 0.017 | 0.071 | 0.38 | 2.906 | 2.162 | 123.8 | | |
| **20↔1\*\*$_{C2'H2}$** | | | | | | | | | | |
| **20** | O5H . . .O4 | 8.38 | 0.048 | 0.136 | 1.36 | 2.600 | 1.700 | 148.6 | 135.5 | 3.62 |
| | O3H . . .C2' | 2.36\* | 0.012 | 0.045 | 251.5 | 3.033 | 2.242 | 138.5 | | |
| **TS$_{20↔1^{**}C2'H2}$** | O5H . . .O4 | 8.71 | 0.049 | 0.138 | 1.11 | 2.591 | 1.692 | 148.5 | 148.4 | 4.07 |
| | HC2'H . . .O3 | 27.22\*\* | 0.111 | 0.085 | 3.56 | 2.509 | 1.398 | 141.0 | | |

(*Continued*)

**Table 2.** (Continued)

| Conformer, TS and tautomer | AH···B H-bond / A···B vdW contact | $E_{AH···B}/E_{A··B}$ [a] | $\rho$ [b] | $\Delta\rho$ [c] | $100·\varepsilon$ [d] | $d_{A···B}$ [e] | $d_{H···B}$ [f] | ∠AH···B [g] | ΔC3-C2-C1'-C6' [h] | μ [i] |
|---|---|---|---|---|---|---|---|---|---|---|
| $1^{**}_{C2'H2}$ [32] | O5H...O4 | 8.62 | 0.049 | 0.138 | 0.89 | 2.594 | 1.697 | 148.2 | 180.0 | 3.40 |
| | C2'...O3 | 4.47* | 0.018 | 0.075 | 458.16 | 2.695 | - | - | | |
| | | | | | $10↔1^*_{C5'H2}$ | | | | | | |
| $TS_{10↔1^*C5'H2}$ | O5H...O4 | 6.47 | 0.040 | 0.124 | 1.48 | 2.657 | 1.774 | 147.1 | -175.7 | 3.21 |
| | O3H...O4 | 3.36 | 0.027 | 0.104 | 57.17 | 2.621 | 2.003 | 119.1 | | |
| | C2'H...O3 | 4.28* | 0.019 | 0.080 | 2.57 | 2.885 | 2.107 | 126.5 | | |
| | HC5'H...O4' | 27.47** | 0.112 | 0.070 | 25.78 | 2.232 | 1.407 | 104.6 | | |
| $1^*_{C5'H2}$ [32] | O5H...O4 | 6.47 | 0.040 | 0.123 | 1.37 | 2.661 | 1.780 | 146.8 | -173.4 | 4.69 |
| | O3H...O4 | 3.36 | 0.027 | 0.106 | 52.39 | 2.614 | 1.994 | 119.2 | | |
| | C5'H...O4' | 4.11* | 0.019 | 0.077 | 3.93 | 2.894 | 2.125 | 125.8 | | |
| **(d) Transition of the proton from the O3H group to the O4 oxygen atom** | | | | | | | | | | |
| | | | | | $1↔1^*_{O5H/O4H}$ | | | | | | |
| $TS_{1↔1^*O5H/O4H}$ | O5H...O4 | 2.65 | 0.024 | 0.090 | 0.90 | 2.802 | 1.964 | 143.0 | 180.0 | 3.79 |
| | O4H...O3 | 22.48 | 0.107 | 0.082 | 1.04 | 2.354 | 1.408 | 136.1 | | |
| | C2'H...O3 | 3.75* | 0.018 | 0.066 | 3.88 | 2.958 | 2.175 | 127.2 | | |
| $1^*_{O5H/O4H}$ [32] | O5H...O4 | 2.89 | 0.025 | 0.100 | 4.07 | 2.754 | 1.923 | 142.1 | 180.0 | 4.06 |
| | O4H...O3 | 7.43 | 0.044 | 0.128 | 12.54 | 2.503 | 1.783 | 125.2 | | |
| | C2'H...O3 | 4.53* | 0.021 | 0.077 | 3.16 | 2.903 | 2.114 | 127.3 | | |
| **(e) Transition of the proton from the O7H group to the C6 carbon atom of the C6H group** | | | | | | | | | | |
| | | | | | $1^{**}_{O5H/O4H/O3H}↔1^*_{O5H/O4H/O3H}$ | | | | | | |
| $1^{**}_{O5H/O4H/O3H}$ [32] | O4H...O5 | 5.04 | 0.034 | 0.128 | 6.17 | 2.634 | 1.806 | 140.7 | 180.0 | 8.58 |
| | O3H...O4 | 2.17 | 0.022 | 0.100 | 66.71 | 2.613 | 2.047 | 115.4 | | |
| | C2'H...O3 | 4.15* | 0.019 | 0.078 | 1.36 | 2.868 | 2.130 | 123.2 | | |
| $TS_{1^{**}O5H/O4H/O3H↔1^*O5H/O4H/O3H}$ | O4H...O5 | 5.75 | 0.037 | 0.135 | 6.00 | 2.613 | 1.765 | 142.5 | 180.0 | 7.99 |
| | O3H...O4 | 2.17 | 0.022 | 0.099 | 79.70 | 2.617 | 2.055 | 115.1 | | |
| | C2'H...O3 | 4.11* | 0.019 | 0.077 | 1.28 | 2.871 | 2.133 | 123.2 | | |
| $1^*_{O5H/O4H/O3H}$ [32] | O4H...O5 | 6.47 | 0.040 | 0.138 | 5.72 | 2.599 | 1.740 | 143.4 | 180.0 | 6.68 |
| | O3H...O4 | 1.93 | 0.021 | 0.097 | 126.99 | 2.633 | 2.076 | 114.7 | | |
| | C2'H...O3 | 4.21* | 0.019 | 0.079 | 1.29 | 2.863 | 2.124 | 123.3 | | |
| | | | | | $1↔1^*_{O5H/O4H/O3H}$ | | | | | | |
| $TS_{1^{**}O4H/O3H↔1^*O5H/O4H/O3H}$ | O4H...O5 | 4.47 | 0.032 | 0.100 | 1.67 | 2.735 | 1.856 | 145.7 | 180.0 | 6.05 |
| | C2'H...O3 | 4.32* | 0.019 | 0.081 | 1.39 | 2.686 | 2.143 | 113.9 | | |
| $1^*_{O5H/O4H/O3H}$ [32] | O4H...O5 | 6.47 | 0.040 | 0.138 | 5.72 | 2.599 | 1.740 | 143.4 | 180.0 | 6.68 |
| | O3H...O4 | 1.93 | 0.021 | 0.097 | 126.99 | 2.633 | 2.076 | 114.7 | | |
| | C2'H...O3 | 4.21* | 0.019 | 0.079 | 1.29 | 2.863 | 2.124 | 123.3 | | |

[a]The energy of the AH···B / A···B specific contact, calculated by Espinose-Molins-Lecomte [94, 95] (marked with an asterisk), Brovarets-Yurenko-Hovorun [96] (marked with a double asterisk) or Nikolaienko-Bulavin-Hovorun [100] formulas, kcal·mol$^{-1}$

[b]The electron density at the (3,-1) BCP of the specific contact, a.u.

[c]The Laplacian of the electron density at the (3,-1) BCP of the specific contact, a.u.

[d]The ellipticity at the (3,-1) BCP of the specific contact

[e]The distance between the A and B atoms of the AH···B / A···B specific contact, Å

[f]The distance between the H and B atoms of the AH···B H-bond, Å

[g]The H-bond angle, degree

[h]The dihedral angle ∠C3-C2-C1′-C6′, degree

[i]The dipole moment of the molecule, Debay. See also Fig 1 and Table 1.

energies of activation consist ~93–96 kcal·mol$^{-1}$ for the **4↔4$^*$$_{O1H}$**, **7↔7$^*$$_{O1H}$** and **10↔10$^*$$_{O1H}$** tautomeric transformations of the non-planar conformers **4**, **7** and **10** (see Fig 1 and Table 1).

At this, the tautomer **4$^*$$_{O1H}$** has been established to be dynamically-unstable (ΔΔG = 0.9 kcal·mol$^{-1}$)–its lifetime τ = 3·10$^{-13}$ s (Table 1) is less than the period of the most low-frequency torsional vibration of the rings around the C2-C1′ bond, which could not develop during this lifetime.

We have also tried to localize the tautomer with the proton, transferred to the O1 oxygen atom from the other neighboring C6H group for others conformers of the quercetin molecule [26] in the case, when these groups are closely located. However, since the stable structure could not be localized, that means that in fact this reaction would not occur.

So, intramolecular proton transfer from the C8H group to O1 oxygen atom causes decyclization (opening) of the C ring of the quercetin molecule. We consider this result quite important, taking into account how much attention attracts prototropic, in particular ring-chain tautomerism [107, 108], in the modern computer-aided drug design [42, 43].

**b) Transition of the proton from the O7H/O3′H hydroxyl groups to the carbon atoms of the neighboring C6H/C2′H groups.** Firstly, we have considered all possible sites for the proton transfer from the hydroxyl groups to the carbon atoms of the neighboring CH groups with the formation of the CH$_2$ group. It was revealed only two tautomerization reactions, which occur in this case–O7H→C6H and O3′H→C2′H. Investigated tautomeric transformations– **1↔1$^*$$_{C6H2}$** (ΔΔG$_{TS}$ = 65.30) and **1↔1$^*$$_{C2'H2}$** (ΔΔG$_{TS}$ = 68.15 kcal·mol$^{-1}$)–occur *via* the intramolecular proton transfer, which are preceded by the rotations of the hydroxyl groups to the CH groups, with Gibbs free energy barriers of activation– 65.30 and 68.15 kcal·mol$^{-1}$, respectively. As a result of these tautomerisations, the planar tautomers **1$^*$$_{C6H2}$** and **1$^*$$_{C2'H2'}$** with relative Gibbs free energies 44.50 and 50.38 kcal·mol$^{-1}$, containing the C6H$_2$ and C2′H$_2$ groups have been formed, respectively (Fig 1, Tables 1 and 2).

These processes of tautomerisation are assisted by the strong intramolecular HC6H . . .O7 (28.46) and HC2′H . . .O3' (26.97 kcal·mol$^{-1}$) H-bonds at the TS$_{1↔1^*C6H2}$ and TS$_{1↔1^*C2'H2}$ transition states. All others H-bonds (O5H . . .O4, O3H . . .O4 and C2′H . . .O3) remain the same at the starting **1** and terminal **1$^*$$_{C6H2}$** structures for the transformation **1↔1$^*$$_{C6H2}$**, while the initial set of the H-bonds (O5H . . .O4, O3H . . .O4, C2′H . . .O3) rearranges into the terminal network of the H-bonds (O5H . . .O4, O3H . . .O4, O3 . . .C2', O4′H . . .O3') for the transformation **1↔1$^*$$_{C2'H2}$** (see Fig 1 and Table 2).

**c) Transitions of the proton from the O7H/O5H/O3H/O4′H hydroxyl groups to the C8/C6/C2′/C5′ carbon atoms of the C8H/C6H/C2′H/C5′H groups, which are preceded by the rotations of the hydroxyl groups around the C7O7/C5O5/C3O3/C4′O4′ bonds by 180 degree.** We have also surveyed other sites of the proton attachment for the possibility of the proton transfer to them. However, analysed sites require rotation of the OH hydroxyl groups around the C-O bond by 180 degree, leading to the prototropic transformations–O7H→C8H, O5H→C6H, O3H→C2′H and O4′H→C5′H. Only in this way of the initial rotation of the OH hydroxyl group of the basic tautomer **1** of the quercetin molecule [26], it is possible to form new prototropic tautomers through the intramolecular transfer of single proton. However, precise investigation of the transformations *via* the rotations of the OH hydroxyl groups would be the subject of the next study [27], since in this paper we are focusing exactly on the mechanisms of the intramolecular proton transfer.

Thus, it was revealed the following chains of the SPT reactions (Fig 1, Table 1): **5↔5$^*$$_{C8H2}$** (ΔΔG$_{TS}$ = 20.46); **25↔25$^*$$_{C6H2}$** (ΔΔG$_{TS}$ = 61.63); **20↔1$^{**}$$_{C2'H2}$** (ΔΔG$_{TS}$ = 34.71) and **10↔1$^*$$_{C5'H2}$** (ΔΔG$_{TS}$ = 70.59 kcal·mol$^{-1}$).

Notably, all of these reactions are assisted by the formation at the TSs of the extremely strong intramolecular HCH . . .O H-bond (26.97–31.44 kcal·mol$^{-1}$ (Table 2)) between the CH$_2$

group and neighboring oxygen atom. At this, all other H-bonds remain practically unchanged at the initial and terminal states (Fig 1, Table 2). Prototropic tautomers, which are formed in this case, are planar structures (Table 2).

Notably, activation barriers for the considered $\mathbf{5}\leftrightarrow\mathbf{5^*}_{\mathbf{C8H2}}$, $\mathbf{25}\leftrightarrow\mathbf{25^*}_{\mathbf{C6H2}}$ and $\mathbf{10}\leftrightarrow\mathbf{1^*}_{\mathbf{C5'H2}}$ tautomerisations are quite high (~62–71 kcal·mol$^{-1}$), except the cases $\mathbf{5}\leftrightarrow\mathbf{5^*}_{\mathbf{C8H2}}$ ($\Delta G = 20.46$) and $\mathbf{20}\leftrightarrow\mathbf{1^{**}}_{\mathbf{C2'H2}}$ ($\Delta G = 34.71$ kcal·mol$^{-1}$). This relatively small value of the barrier can be explained by the formation of the six-membered ring at the $TS_{20\leftrightarrow1^{**}C2'H2}$ and by moving of the proton along the O3H...C2′ H-bond [26]. In those cases, when $TS_{25\leftrightarrow25^*C6H2}$, $TS_{5\leftrightarrow5^*C8H2}$ and $TS_{10\leftrightarrow1^*C5'H2}$ contains four-membered rings and proton does not move along the intramolecular H-bond–the values of the activation barriers are much higher. At this, $\mathbf{1^{**}}_{\mathbf{C2'H2}}$ tautomer is the only one structure, which has the C2 = C1′ double bond.

d) Transition of the proton from the O3H hydroxyl group to the O4 atom.

Further we investigated structural mechanisms of the single proton transfer, occurring between the O5H and O3H hydroxyl groups. Thus, it was found that proton can transfer from the O3H hydroxyl group to the O4 oxygen atom through the $\mathbf{1}\leftrightarrow\mathbf{1^*}_{\mathbf{O5H/O4H}}$ tautomerization reaction with the barrier $\Delta\Delta G_{TS} = 13.10$ kcal·mol$^{-1}$. However, terminal localized complex is dynamically unstable–reverse Gibbs free energy barrier has negative value ($\Delta\Delta G = -1.20$ kcal·mol$^{-1}$) (exactly in this case it is observed at TSs the lowest value of the imaginary frequency $\nu_i = 892.1$ cm$^{-1}$) (Table 1).

It is logically to think by analogy that the same intramolecular proton transfer should occur from the O5H hydroxyl group to the O4 oxygen atom. But in this case the TSs and tautomers could not be localized at all.

e) Proton migration from the O7H/O5H hydroxyl groups to the C6 atom of the C6H group.

We also considered tautomeric transformation of the $\mathbf{1^*}_{\mathbf{O5H/O4H/O3H}}$ tautomer by the transition of the protons from the O7H/O5H hydroxyl groups to the neighboring C6 atom of the C6H group.

Thus, in the first case the $\mathbf{1^{**}}_{\mathbf{O5H/O4H/O3H}}\leftrightarrow\mathbf{1^*}_{\mathbf{O5H/O4H/O3H}}$ tautomerization reaction proceeds *via* the transfer of proton from the O7H hydroxyl group to the neighboring C6 atom and occurs *via* the quite high barrier ($\Delta\Delta G_{TS} = 49.01$ kcal·mol$^{-1}$) and leads to the dynamically stable tautomer $\mathbf{1^*}_{\mathbf{O5H/O4H/O3H}}$ (Fig 1, Table 1).

In the second case, the intramolecular proton transfer in the $\mathbf{1^{**}}_{\mathbf{O4H/O3H}}$ tautomer from the O5H hydroxyl group to the neighboring C6 carbon atom of the C6H group– $\mathbf{1^{**}}_{\mathbf{O4H/O3H}}\leftrightarrow\mathbf{1^*}_{\mathbf{O5H/O4H/O3H}}$ –occurs through the $TS_{1^{**}O4H/O3H\leftrightarrow1^*O5H/O4H/O3H}$ ($\Delta\Delta G_{TS} = 75.24$ kcal·mol$^{-1}$) and leads to the formation of the dynamically unstable $\mathbf{1^{**}}_{\mathbf{O4H/O3H}}$ tautomer with relative electronic energy 14.87 kcal·mol$^{-1}$, which further causes chain transfer of the proton from the O4H hydroxyl group to the O5 oxygen atom, leading to the stable conformer $\mathbf{1}$ (Fig 1, Table 1).

It can be expected the reduction of the values of the activation barriers at the consideration of these transitions in the polar solutions or assisted by various ligands.

## Conclusions and perspectives

Presented QM/QTAIM computational modeling of the tautomers formation through the intramolecular proton transfer shows that the quercetin molecule is able to tautomerise *via* the different routes within the framework of the classical valency rules:

a. Proton transfer from the C8H group to the O1 atom, leading in three cases to the breakage of the C ring: $\mathbf{1}\leftrightarrow\mathbf{1^*}_{\mathbf{O1H}}$, $\mathbf{7}\leftrightarrow\mathbf{7^*}_{\mathbf{O1H}}$ and $\mathbf{10}\leftrightarrow\mathbf{10^*}_{\mathbf{O1H}}$, except the case of $\mathbf{4}\leftrightarrow\mathbf{4^*}_{\mathbf{O1H}}$ reaction ($\Delta\Delta G_{TS} \sim 93$–$96$ kcal·mol$^{-1}$).

b. Transition of the proton from the O7H/O3′H hydroxyl groups to the carbon atoms of the neighboring C6H/C2′H groups: $1\leftrightarrow1^*_{C6H2}$ and $1\leftrightarrow1^*_{C2'H2}$ ($\Delta\Delta G_{TS}$ ~ 65–68 kcal·mol$^{-1}$).

c. Migration of the proton from the O7H/O5H/O3H/O4′H hydroxyl groups to the carbon atoms of the C8H/C6H/C2′H/C5′H groups, preceded by the rotations of the hydroxyl groups around the C7O7/C5O5/C3O3/C4′O4′ bond by 180 degree: $5\leftrightarrow5^*_{C8H2}$ ($\Delta\Delta G_{TS}$ = 20.46); $25\leftrightarrow25^*_{C6H2}$ ($\Delta\Delta G_{TS}$ = 61.63); $20\leftrightarrow1^{**}_{C2'H2}$ ($\Delta\Delta G_{TS}$ = 34.71) and $10\leftrightarrow1^*_{C5'H2}$ ($\Delta\Delta G_{TS}$ = 70.59 kcal·mol$^{-1}$).

d. Proton transfer from the O3H hydroxyl group to the O4 oxygen atom with the formation of the dynamically-unstable tautomer: $1\leftrightarrow1^*_{O5H/O4H}$ ($\Delta\Delta G_{TS}$ ~ 13 kcal·mol$^{-1}$).

e. Transition of the proton from the O7H/O5H hydroxyl group to the C6 carbon atom of the C6H group: $1^{**}_{O5H/O4H/O3H}\leftrightarrow1^*_{O5H/O4H/O3H}$ and $1^{**}_{O4H/O3H}\leftrightarrow1^*_{O5H/O4H/O3H}$ ($\Delta\Delta G_{TS}$ ~ 49–75 kcal·mol$^{-1}$).

These prototropic transformations of the quercetin molecule are accompanied by the geometrical changes, dipole moment rearrangement and breakage or formation of the intramolecular specific contacts (H-bonds and attractive van der Waals contacts).

It was demonstrated that the most probable process among all investigated is the proton transfer from the O3H hydroxyl group to the C2′ carbon atom of the C2′H of the B ring along the intramolecular O3H...C2′ H-bond with the further formation of the C2′H$_2$ group, while the least probable proton transfer occurs from the C8H group to the O1 oxygen atom–causes the decyclization of the C ring.

Obtained results can be useful for the planning of targeted chemical experiments, aimed at the acceleration of the reaction of intramolecular tautomerization of a quercetin molecule by the ligands of different structure and origin, as well as for the better understanding of the mechanisms of the course of reactions, related to the metabolism of the quercetin molecule.

## Acknowledgments

The authors gratefully appreciate technical support and computational facilities of joint computer cluster of SSI "Institute for Single Crystals" of the National Academy of Sciences of Ukraine (NASU) and Institute for Scintillation Materials of the NASU incorporated into Ukrainian National Grid.

DrSci Ol'ha O. Brovarets' expresses sincere gratitude to the U.S.-Ukraine Foundation (USUF) Biotech Initiative for a travel grant ("2018 Emerging Biotech Leader of Ukraine"; https://www.usukraine.org/biotechnology-initiative/), enabling to participate in the "63$^{rd}$ Annual Meeting of the Biophysical Society BPS'2019" (Baltimore, Maryland, March 2–6, 2019; https://www.biophysics.org/2019meeting#/; https://bioukraine.org/news/emerging-biotech-leader-olha-brovarets-attends-63rd-biophysical-society-meeting-in-baltimore/; https://bioukraine.org/news/emerging-leader-olha-brovarets-shares-her-us-experience-with-bionity-student-biotech-club/).

DrSci Ol'ha O. Brovarets' sincerely thanks for the Scholarship of Verkhovna Rada (Parliament) of Ukraine for the talented young scientists given in 2019 year.

## Author Contributions

**Conceptualization:** Ol'ha O. Brovarets', Dmytro M. Hovorun.

**Data curation:** Ol'ha O. Brovarets', Dmytro M. Hovorun.

**Formal analysis:** Ol'ha O. Brovarets', Dmytro M. Hovorun.

**Funding acquisition:** Dmytro M. Hovorun.

**Investigation:** Ol'ha O. Brovarets', Dmytro M. Hovorun.

**Methodology:** Dmytro M. Hovorun.

**Project administration:** Dmytro M. Hovorun.

**Resources:** Dmytro M. Hovorun.

**Software:** Dmytro M. Hovorun.

**Supervision:** Dmytro M. Hovorun.

**Validation:** Dmytro M. Hovorun.

**Visualization:** Ol'ha O. Brovarets', Dmytro M. Hovorun.

**Writing – original draft:** Ol'ha O. Brovarets', Dmytro M. Hovorun.

**Writing – review & editing:** Ol'ha O. Brovarets', Dmytro M. Hovorun.

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
