## [Decision Letter · Decision Letter 0]

1 Oct 2019

PONE-D-19-25268

Intramolecular tautomerization of the quercetin molecule via the proton transfer: QM computational study

PLOS ONE

Dear Prof. Hovorun,

Thank you for submitting your manuscript to PLOS ONE. After careful consideration, we feel that it has merit but does not fully meet PLOS ONE’s publication criteria as it currently stands. Therefore, we invite you to submit a revised version of the manuscript that addresses the points raised during the review process.

Two reviewers have suggested that minor revision is required and I concur.  Your revision must address all of the issues raised by the reviewers by revision and/or rebuttal.

We would appreciate receiving your revised manuscript by Nov 15 2019 11:59PM. To enhance the reproducibility of your results, we recommend that if applicable you deposit your laboratory protocols in protocols.io, where a protocol can be assigned its own identifier (DOI) such that it can be cited independently in the future. For instructions see: http://journals.plos.org/plosone/s/submission-guidelines#loc-laboratory-protocols

We look forward to receiving your revised manuscript.

Kind regards,

Dennis Salahub

Academic Editor

PLOS ONE

Journal Requirements:

1. We noticed you have some minor occurrence of overlapping text with the following previous publication(s), which needs to be addressed:

- https://www.tandfonline.com/doi/abs/10.1080/07391102.2019.1645734?journalCode=tbsd20

-https://www.tandfonline.com/doi/abs/10.1080/07391102.2019.1656671?journalCode=tbsd20

In your revision ensure you cite all your sources (including your own works), and quote or rephrase any duplicated text outside the methods section. Further consideration is dependent on these concerns being addressed.

2. Thank you for including the following funding information within the acknowledgements section of your manuscript; "DrSci Ol’ha O. Brovarets’ expresses sincere gratitude to the U.S.-Ukraine Foundation (USUF) Biotech Initiative for a travel grant (“2018 Emerging Biotech Leader of Ukraine”; https://www.usukraine.org/biotechnology-initiative/), enabling to participate in the “63rd Annual Meeting of the Biophysical Society BPS'2019” (Baltimore, Maryland, March 2-6, 2019; https://www.biophysics.org/2019meeting#/;
https://bioukraine.org/news/emerging-biotech-leader-olhabrovarets-attends-63rd-biophysical-society-meeting-in-baltimore/;
https://bioukraine.org/news/emergingleader-olha-brovarets-shares-her-us-experience-with-bionity-student-biotech-club/). DrSci Ol’ha O. Brovarets’ sincere thanks for the Scholarship of Verkhovna Rada (Parliament) of Ukraine for the talented young scientists given in 2019 year."

Reviewers' comments:

Reviewer's Responses to Questions

**Comments to the Author**

1. Is the manuscript technically sound, and do the data support the conclusions?

Reviewer #1: Yes

Reviewer #2: Yes

2. Has the statistical analysis been performed appropriately and rigorously? 

Reviewer #1: Yes

Reviewer #2: No

3. Have the authors made all data underlying the findings in their manuscript fully available?

Reviewer #1: Yes

Reviewer #2: Yes

4. Is the manuscript presented in an intelligible fashion and written in standard English?

Reviewer #1: Yes

Reviewer #2: Yes

5. Review Comments to the Author

Reviewer #1: I do not accept the first sentence of the abstract "Quercetin molecule (3, 3′, 4′, 5, 7-pentahydroxyflvanone, C15H10O7) is

an important flavonoid compound, containing two aromatic A and B rings linked through the C ring containing oxygen and five OH hydroxyl groups attached to the 3, 3′, 4′, 5 and 7 positions. This molecule is found in many foods and plants, and is known to act as a natural drug molecule with a wide range of treatment properties, like an anti-oxidant, anti-toxic etc."

1) Quercetin is not a natural drug! So far it is not proved! The anti-oxidant, anti-toxic effects are known, but their therapeutic treatments are not proper studied.

2) The standard quercetin is the 3, 3′, 4′, 5, 7-pentahydroxyflvanone, if we use a standard numeration. The author's numeration corresponds to rotation of the B ring by 180o around C2-C1' bond in respect to the standard! Their molecule is

3, 5′, 4′, 5, 7-pentahydroxyflvanone (in respect to the standard numeration) and the choice of the calculated molecule corresponds to rotation of the B ring by 180o around C2-C1' bond axis. May be the conformer used by authors is the most stable one according to their previous studies, but this is unknown for the plain reader when he starts to read the paper. This should be mentioned from the begining.

3) The first part of the sentence is a simple tautology and needs to be removed.

4) Why proton transfer? (not H atom transfer). The corresponding polarization is not determined.

The MS is interesting and continues the previous studies of quercetin isomers. This molecule is of great biological and medical importance; it received great attention during recent years. All DFT optimizations of transition states and global minima are well documented including Bader's QTAIM analysis of bond critical points. The topology of the electron density was analyzed, using program package AIM’2000 with all default options and wave functions obtained at the level of theory

used for geometry optimisation. The presence of the (3,-1) bond critical point (BCP), bond path between hydrogen donor and acceptor and positive value of the Laplacian at this BCP (Δρ>0) were considered altogether as criteria for the formation of the H-bond and attractive van der Waals contacts. Thus, the MS is acceptable after minor revision (including abstract improvement).

The main objection concerns the influence of the solvent on predictions of this study. There are no comments on this important practical question. In page 6 we have: "All calculations were performed for the tautomeric transitions of the quercetin molecule as their intrinsic property, that is adequate for modeling of the processes occurring in real systems".

Calculations of quercetin in vacuum? with proton transfer? This is not a real system. At least, some comments are necessary.

There are some typos. Even in the first sentence of the abstract one reads"pentahydroxyflvanone"? Thus, the careful reading would be useful.

Reviewer #2: The manuscript by Brovarets’ and Hovorun addresses several intramolecular proton transfer pathways in the quercetin molecule. The authors also examined the properties of the transition states related to these transfers. The employed methods are sound and the results can be reproduced. The authors treated the literature correctly. Yet, they should avoid overcitation of their previous work not related to quercetin. In my opinion, the results of this study are not significant to the broader readership of the Journal since possible biological and chemical roles of the examined pathways are not discussed. The manuscript is not well-written and I suggest a revision of the manuscript.

Additional comments:

Page 2, the sentence “It has attracted a lot of attention last time, due to the wide range of its treatment properties, …” should be modified.

Page 4, DNA bases are not related to their work and references 42-66 might be excluded from the manuscript

Page 4, this text is not particularly important for the presented work: “This points on the fact that proton transfer defines the quantum nature of the biological objects, so-called field of quantum biology, which was started as a separate discipline from late 1920s, when Niels Bohr, delivered an influential lecture on whether the “atomic theory” could help to solve the mystery of life.”

Page 9, “inramolecular proton”

Page 10, “in factthis”

Figure 1, middle panel “hydoxyl”

6. PLOS authors have the option to publish the peer review history of their article (what does this mean?). If published, this will include your full peer review and any attached files.

Reviewer #1: Yes: Boris F. Minaev

Reviewer #2: No

---

## [Author Response · Author response to Decision Letter 0]

17 Oct 2019

Dear Editor, Professor Dennis Salahub!

On behalf of all the co-authors, I would like to express to you sincere appreciation for the providing of the comprehensive reviewing of our manuscript.

We made our best efforts to improve it in the strict accordance with the reviewers’ comments and remarks. Also, we have modified methods section in over to avoid the repetition. 

At this all changes are highlighted in yellow. 

I hope very much to receive your positive decision!

Best regards,

Professor Dmytro Hovorun

Corresponding Author 

Reviewer #1: 

Comment:

I do not accept the first sentence of the abstract "Quercetin molecule (3, 3′, 4′, 5, 7-pentahydroxyflvanone, C15H10O7) is an important flavonoid compound, containing two aromatic A and B rings linked through the C ring containing oxygen and five OH hydroxyl groups attached to the 3, 3′, 4′, 5 and 7 positions. This molecule is found in many foods and plants, and is known to act as a natural drug molecule with a wide range of treatment properties, like an anti-oxidant, anti-toxic etc." 

1) Quercetin is not a natural drug! So far it is not proved! The anti-oxidant, anti-toxic effects are known, but their therapeutic treatments are not proper studied. 

2) The standard quercetin is the 3, 3′, 4′, 5, 7-pentahydroxyflvanone, if we use a standard numeration. The author's numeration corresponds to rotation of the B ring by 180o around C2-C1' bond in respect to the standard! Their molecule is 3, 5′, 4′, 5, 7-pentahydroxyflvanone (in respect to the standard numeration) and the choice of the calculated molecule corresponds to rotation of the B ring by 180o around C2-C1' bond axis. May be the conformer used by authors is the most stable one according to their previous studies, but this is unknown for the plain reader when he starts to read the paper. This should be mentioned from the begining. 

3) The first part of the sentence is a simple tautology and needs to be removed. 

Reply:

We would like to sincerely thank to Reviewer for the comprehensive reviewing of our manuscript and performed comments. Below we provided the answers. 

1) Using this statement we have been based on the recent literature data according quercetin. We have also added more reviews and overviews of patents on the studies according the action of quercetin. We agree with Reviewer that more studies are needed to better characterize the mechanisms of theraupeutic action of quercetin.

2), 3) We have corrected this sentence, at this a commonly used designation of quercetin has been used. 

Comment:

4) Why proton transfer? (not H atom transfer). The corresponding polarization is not determined. 

Reply:

In this paper we use the generally accepted in the literature term “proton transfer”. By default, it is considered that proton transfer would cause reorganization of the electronic density of the molecule. 

Comment:

The MS is interesting and continues the previous studies of quercetin isomers. This molecule is of great biological and medical importance; it received great attention during recent years. All DFT optimizations of transition states and global minima are well documented including Bader's QTAIM analysis of bond critical points. The topology of the electron density was analyzed, using program package AIM’2000 with all default options and wave functions obtained at the level of theory used for geometry optimisation. The presence of the (3,-1) bond critical point (BCP), bond path between hydrogen donor and acceptor and positive value of the Laplacian at this BCP (Δρ>0) were considered altogether as criteria for the formation of the H-bond and attractive van der Waals contacts. Thus, the MS is acceptable after minor revision (including abstract improvement). 

Reply:

Thank you for the comprehensive analysis and provided comments. 

Comment:

The main objection concerns the influence of the solvent on predictions of this study. There are no comments on this important practical question. In page 6 we have: "All calculations were performed for the tautomeric transitions of the quercetin molecule as their intrinsic property, that is adequate for modeling of the processes occurring in real systems". 

Calculations of quercetin in vacuum? with proton transfer? This is not a real system. At least, some comments are necessary. 

Reply:

The aim of the present study was to investigate the intrinsically inherent properties of the quercetin molecule. Further, based on these studies on the basic properties of the quercetin molecule it would be possible to investigate more complex processes and systems, taking into account different ligands or specific environments, which could possibly accelerate these tautomerization processes. 

Comment:

There are some typos. Even in the first sentence of the abstract one reads"pentahydroxyflvanone"? Thus, the careful reading would be useful. 

Reply: We have removed this typos and proofread manuscript once more.

Reviewer #2: 

Comment: 

The manuscript by Brovarets’ and Hovorun addresses several intramolecular proton transfer pathways in the quercetin molecule. The authors also examined the properties of the transition states related to these transfers. The employed methods are sound and the results can be reproduced. The authors treated the literature correctly. Yet, they should avoid overcitation of their previous work not related to quercetin. In my opinion, the results of this study are not significant to the broader readership of the Journal since possible biological and chemical roles of the examined pathways are not discussed. The manuscript is not well-written and I suggest a revision of the manuscript. 

Reply:

We are thankful to reviewer for the comprehensive analysis of our manuscript and also for the critical remarks, which we have implemented at the revision of the manuscript. 

Comment: 

Additional comments: 

Page 2, the sentence “It has attracted a lot of attention last time, due to the wide range of its treatment properties, …” should be modified. 

Page 4, DNA bases are not related to their work and references 42-66 might be excluded from the manuscript 

Page 4, this text is not particularly important for the presented work: “This points on the fact that proton transfer defines the quantum nature of the biological objects, so-called field of quantum biology, which was started as a separate discipline from late 1920s, when Niels Bohr, delivered an influential lecture on whether the “atomic theory” could help to solve the mystery of life.” 

Page 9, “inramolecular proton” 

Page 10, “in factthis” 

Figure 1, middle panel “hydoxyl” 

Reply:

We have modified provided sentences and phrases.

---

## [Decision Letter · Decision Letter 1]

22 Oct 2019

Intramolecular tautomerization of the quercetin molecule due to the proton transfer: QM computational study

PONE-D-19-25268R1

Dear Dr. Hovorun,

We are pleased to inform you that your manuscript has been judged scientifically suitable for publication and will be formally accepted for publication once it complies with all outstanding technical requirements.

With kind regards,

Dennis Salahub

Academic Editor

PLOS ONE

Additional Editor Comments (optional):

Reviewers' comments:

Reviewer's Responses to Questions

**Comments to the Author**

1. If the authors have adequately addressed your comments raised in a previous round of review and you feel that this manuscript is now acceptable for publication, you may indicate that here to bypass the “Comments to the Author” section, enter your conflict of interest statement in the “Confidential to Editor” section, and submit your "Accept" recommendation.

Reviewer #1: All comments have been addressed

Reviewer #2: All comments have been addressed

2. Is the manuscript technically sound, and do the data support the conclusions?

Reviewer #1: Yes

Reviewer #2: Yes

3. Has the statistical analysis been performed appropriately and rigorously? 

Reviewer #1: Yes

Reviewer #2: Yes

4. Have the authors made all data underlying the findings in their manuscript fully available?

Reviewer #1: Yes

Reviewer #2: Yes

5. Is the manuscript presented in an intelligible fashion and written in standard English?

Reviewer #1: Yes

Reviewer #2: Yes

6. Review Comments to the Author

Reviewer #1: As far as solvent account concens you refer to future elaboration. I should prefer to write short comment in your present version about limitation of the used vacuum approach.

This is optional deal.

Reviewer #2: I am satisfied with the corrections and thus recommend the manuscript for the publication in the present form.

7. PLOS authors have the option to publish the peer review history of their article (what does this mean?). If published, this will include your full peer review and any attached files.

Reviewer #1: Yes: Boris F. Minaev

Reviewer #2: No

---

## [Editor Report · Acceptance letter]

12 Nov 2019

PONE-D-19-25268R1 

Intramolecular tautomerization of the quercetin molecule due to the proton transfer: QM computational study 

Dear Dr. Hovorun:

I am pleased to inform you that your manuscript has been deemed suitable for publication in PLOS ONE. Congratulations! Your manuscript is now with our production department. 

With kind regards,

on behalf of

Dr. Dennis Salahub 

Academic Editor

PLOS ONE